# Transgression Related Holocene Coastal Glendonites from Historic Sites

**Bo Schultz** [1,*] , **Jennifer Huggett** [2] , **Bas van de Schootbrugge** [3] , **Clemens V. Ullmann** [4] **and Mathias C. Broch** [1]

1   Museum Salling, Fur Museum, Nederby 28, 7884 Fur, Denmark; broc@museumsalling.dk
2   The Natural History Museum, Cromwell Road, London SW7 5BD, UK; jmhuggett@icloud.com
3   Utrecht University, Princetonlaan 8A, 3584 CS Utrecht, The Netherlands; b.vanderschootbrugge@uu.nl
4   Cambourne School of Mines, University of Exeter, Penryn Campus, Penryn TR10 9FE, UK;
    c.ullmann@exeter.ac.uk
*   Correspondence: bosc@museumsalling.dk

**Abstract:** This study examines the occurrence of glendonite along coastlines since 1825, which have been previously referred to under different names such as Pseudogaylussite, Fundylite, and Kool Hoot across eleven sites. By utilising element ratios and $^{14}$C radiometric dating techniques, we establish a more accurate chronology for these varied sites ranging from 10 to 1 thousand years before the present (Ky BP). Sites include tidal flats, coastal barrier islands, and Wadden Sea environments. While some sites still exist, others are only known through publications and museum collections. Our research expands upon previous findings by presenting petrographic evidence that correlates with glendonite formation. Through the examination of the Olenitsa site on the Kola Peninsula, we demonstrate that marine bioclasts enclosed within concretions surrounding glendonites provide temporal context, suggesting that these outcrops were formed during a single event under changing conditions. Notably, certain sediment structures at selected sites indicate the occurrence of cold-water ice-raft storm events and the presence of drop stones. Furthermore, our paper explores the association of historic coastal sites with the formation of ikaite, highlighting the limitations of relying solely on geochemistry and isotopic analysis for interpretation. Intriguingly, we observe that pseudomorphs are abundant in specific areas but absent in adjacent regions with similar environmental, physical, and chemical conditions. No apparent connection is found between volcanic dust cloud-induced cold spells and glendonite. The distribution of coastal glendonites is more likely related to periods of climatic cooling through other means. We show that radiometric dating with $^{14}$C provides an indication of age, but the results can be erroneous due to the inclusion of older carbon sources in the analysis. The oldest locations discussed in this study are Kool Hoot (Alaska) and the river Clyde (Scotland), and the youngest glendonites discussed are from the Bay of Fundy in Canada. Occurrences from the Wadden Sea are intermediate in age and sit between the other two groups. The age of the Olenitsa site on the Russian Kola Peninsula is uncertain and still debated. We show that measuring the ratio of Mg/Ca can indicate how much the recrystallised ikaite preserved as calcite is influenced by diagenetic pore waters.

**Keywords:** Pseudogaylussite; Glendonite; Ikaite





## 1. Introduction

The first scientific description of ikaite was made in June 1963 at Ikka Fjord in Greenland by Professor Hans Pauly [1]. Ikaite is a metastable mineral with the chemical formula ($CaCO_3 \cdot 6H_2O$) and breaks down above ~7 °C [2,3]. This Ikka Fjord ikaite was found in the form of large, sub-merged tufa pillars, bearing no resemblance to glendonite [2,3]. Latest research shows that ikaite columns in Ikka Fjord form where the seawater is slowly heating up, with the result that the ikaite transforms into monohydrocalcite and aragonite, rather than calcite, as is the norm with glendonite [4]. When ikaite transforms to calcite in cave

systems, it can leave behind only a few calcite preserved ($CaCO_3$) structural "relicts" [5], often in the form of zoned mm large calcite grains, termed "guttulatic", meaning little drop" [6]. Guttulatic crystals have pseudo-hexagonal or spherical cores with low Mg cores [7] formed during the recrystallisation of marine ikaite and are particularly dominant in glendonite [5,6,8–15]. However, guttalatic structure is present in many different aquatic and terrestrial systems. Especially a very defining common feature for glendonite derived for marine settings [15]. In 1982, Professor Erwin Suess made a significant observation when he retrieved large yellow/orange euhedral ikaite crystals from the seabed in Bransfield Strait, offshore Antarctica (Antarctic Core 1138-4) [8]. A striking similarity between ikaite and glendonites was observed, and it was tentatively concluded that ikaite is the precursor mineral of glendonite [9]. Glendonite can only have ikaite as a precursor mineral [9,10]. In marine-influenced sediments, euhedral ikaite appears to be stable above ~7 °C. Ikaite is highly unstable and rapidly breaks down into wet calcite granules within hours of retrieval [11], which has hindered traditional crystallographic investigations.

Historically, while White Sea hornlets had long been identified as glendonite, it was only with ikaite discoveries such as at Bransfield Strait that glendonite ($CaCO_3$) was gradually accepted as a pseudomorph after the metastable mineral ikaite ($CaCO_3*6H_2O$) [12,13]. The narrow range of isotopic compositions of ikaite indicates that precipitation occurs only in specific porewater geochemistry, with the crystal growth halting once the conditions shift due to precipitation or other processes, according to [14]. Recent work on the ikaite from Bransfield Strait concludes that precipitation is restricted to the zone of hydrogenotrophic methanogenesis between the sulphate reduction zone and the deeper zone of methanogenesis [14]. Marine sedimentary ikaite crystals are frequently precipitated in one event with few traces of secondary growth.

Glendonite is often found within enclosing concretions. However, for pre-Holocene pseudomorphs and their concretions, dating techniques lack the precision to distinguish between the ages of the two cementation intervals. Although it is now widely accepted and shown that ikaite is the precursor mineral to glendonite in preserved pseudomorphs, much remains to be learned about glendonite genesis [11,16,17]. The exact conditions necessary for ikaite precipitation are still elusive, but there is an association with freshwater seepage into marine or saline waters, as observed at Ikka Fjord [2–4] and Mono Lake [18].

Ikaite in tundra or tidal coastal areas can be associated with the transgression of marine waters through tidal or storm action [7]. The [14]C dating results from the investigated sites also indicate that this phenomenon occurred during the later stages of postglacial sea level rise. However, it should be noted that the preserved sites represent coastal areas, which are often high-energy environments influenced by storms and tidal floods and represent erosive environments with high rates of re-deposition.

Previous studies have reported data on single-source sediment containing diagenetic carbonate in fully marine glendonites [10,12,14]. Here, we provide a historical context in Figure 1 to coastal ikaite formation in, describe the morphology and petrology in Figure 2, and present new [14]C dating and stable isotope in analysis of carbon and oxygen for the described glendonites in Table 1. Combining with Ca and Mg ration from Table 2, we add good data to future work on a unifying theory of ikaite precipitation.

**Table 1.** $^{14}$C stable isotope results.

| Country | Locality | Figure | Site | Sample | AMS spl. | $^{14}$C age yr BP | $\delta^{13}$C ‰ V-PDB AMS | $\delta^{18}$O ‰ V-PDB |
|---|---|---|---|---|---|---|---|---|
| Canada | Nova Scotia, Bay of Fundy, Sackville | 2(a1) | site II | Pseudomorph | 31880 | 1221 ± 26 | −13 ± 1 | |
| Canada | Nova Scotia, Bay of Fundy, Sackville | 2(a2) | site I | Pseudomorph | 10320 | 1046 ± 22 | −27.87 | −3.38 |
| Canada | Nova Scotia, Bay of Fundy, Sackville | 2(a2) | site I | Pseudomorph | 33804 | 1059 ± 27 | −11 ± 1 | |
| Canada | Nova Scotia, Bay of Fundy, Sackville | 2(a3) | site I | Concretion | 33805 | 762 ± 30 | −12 ± 1 | |
| USA | Alaska, Utqiagvik (Barrow), Isatkoak Lagune ** | | | Pseudomorph | 28500 | 1939 ± 28 | −18 | |
| USA | Alaska, Utqiagvik (Barrow), Isatkoak Lagune ‡ | | | Ikaite | | | −19.8 | −9.9 |
| USA | Alaska, Utqiagvik (Barrow), Isatkoak Lagune ‡ | | | Ikaite | | | −19.7 | |
| Germany | Nordfriesland, Wadden Sea | 2(b2) | Skt. Peter-Ording | Pseudomorph | | | −18.04 | −1.75 |
| Germany | Nordfriesland, Wadden Sea | 2(b2) | Skt. Peter-Ording | Pseudomorph | 28504 | 3676 ± 30 | −16 ± 1 | |
| Germany | Nordfriesland, Wadden Sea | 2(b2) | Halligen/Südfall | Pseudomorph | | | −18.58 | −1.08 |
| Germany | Nordfriesland, Wadden Sea | 2(b2) | Halligen/Südfall | Pseudomorph | 29578 | 3666 ± 35 | −13 ± 1 | |
| Germany | Nordfriesland, Wadden Sea | 2(b2) | Kating | Pseudomorph | 28502 | 2872 ± 40 | −17 ± 1 | |
| Germany | Nordfriesland, Wadden Sea | 2(b2) | Eiderstadt | Pseudomorph | 29577 | 2911 ± 22 | −15 ± 1 | |
| Germany | Nordfriesland, Wadden Sea | 2(b2) | Tönning | Pseudomorph | 34821 | 3209 ± 25 | −12 ± 1 | |
| Germany | Nordfriesland, Wadden Sea | 2(b2) | Tönning | Pseudomorph | | | −13.47 | −1.06 |
| Germany | Nordfriesland, Wadden Sea | 2(b2) | Jever | Pseudomorph | 29579 | 2133 ± 26 | −14 ± 1 | |
| UK | Scotland, Glasgow, River Clyde estuary, Cardross mudflats | 2(c2) | | Snail | 18527 | 4978 ± 25 | 2.49 ± 0.05 | 1.43 ± 0.05 |
| UK | Scotland, Glasgow, River Clyde estuary, Cardross mudflats | 2(c2) | | Pseudomorph | 34820 | 5693 ± 31 | −29 ± 1 | |
| UK | Scotland, Glasgow, River Clyde estuary, Cardross mudflats | 2(c2) | | Concretion | 34821 | 5706 ± 33 | −25 ± 1 | |
| UK | Scotland, Glasgow, River Clyde estuary, Cardross mudflats † | 2(c2) | | Pseudomorph | | | −28.66 | −0.43 |
| UK | Scotland, Glasgow, River Clyde estuary, Cardross mudflats † | 2(c2) | | Pseudomorph | | | −28.24 | −0.57 |
| Netherlands | Noord-Holland, Den Hoorn | 2(b2) | Kwadijk | Pseudomorph | 28503 | 5100 ± 34 | −9 ± 1 | |
| Netherlands | Noord-Holland, Den Hoorn | 2(b2) | Bobeldijk | Pseudomorph | 28501 | 5203 ± 32 | −18 ± 1 | |
| USA | Alaska, Seward Peninsula, Sarichef Island, Shishmaref coast | 2(d1) | | Clam | 34816 | 5278 ± 38 | −2 ± 1 | |
| USA | Alaska, Seward Peninsula, Sarichef Island, Shishmaref coast | 2(d1) | | Pseudomorph | 34817 | 9853 ± 49 | −8 ± 1 | |
| USA | Alaska, Seward Peninsula, Sarichef Island, Shishmaref coast | 2(d1) | | Pseudomorph | 31878 | 10,162 ± 47 | −20 ± 1 | |
| USA | Alaska, Seward Peninsula, Sarichef Island, Shishmaref coast | 2(d1) | | Pseudomorph | 31879 | 9653 ± 43 | −22 ± 1 | |
| USA | Alaska, Seward Peninsula, Sarichef Island, Shishmaref coast | 2(d1) | | Concretion | 34818 | 10,702 ± 46 | 3 ± 1 | |
| Russia | Kola Peninsula, White Sea, Terski coast at Olenitsa * | 2(e1) | | Pseudomorph | | | −14.08 | −1.13 |
| Russia | Kola Peninsula, White Sea, Terski coast at Olenitsa | 2(e1) | | Pseudomorph | 33803 | 8870 ± 61 | −21 ± 1 | |
| Russia | Kola Peninsula, White Sea, Terski coast at Olenitsa * | 2(e1) | | Concretion | | | −10.68 | 0.02 |
| Russia | Kola Peninsula, White Sea, Terski coast at Olenitsa | 2(e1) | | Concretion | 33802 | 8559 ± 52 | −19 ± 1 | |
| Russia | Kola Peninsula, White Sea, Terski coast at Olenitsa * | 2(e1) | | Bivalve | | | −1.22 | 0.05 |
| Russia | Kola Peninsula, White Sea, Terski coast at Olenitsa | 2(e1) | | Bivalve | 33801 | 8835 ± 49 | 1 ± 1 | |
| Russia | Kola Peninsula, White Sea, Terski coast at Olenitsa * | 2(e1) | | Mussel | | | −0.98 | 0.1 |
| Russia | Kola Peninsula, White Sea, Terski coast at Olenitsa | 2(e1) | | Mussel | 10321 | 8485 ± 65 | −1.5 | −4.35 |

** Schultz [10]; ‡ Kennedy [13]; * Published in Vickers [17]; † Published in Shearman and Smith [9].

**Table 2.** Elemental ratios. The full data-sheet can be viewed in supplementary data.

| Country | Site | Figure | Site | Sample | Mg/Ca (mmol/mol) | Sr/Ca (mmol/mol) | Mn/Ca (mmol/mol) | Fe/Ca (mmol/mol) | S/Ca (mmol/mol) | P/Ca (mmol/mol) |
|---|---|---|---|---|---|---|---|---|---|---|
| Canada | Nova Scotia, Bay of Fundy, Sackville | 2(a2) | site II | Pseudomorph | 32.9 | 1.33 | 0.58 | 1.6 | 5.3 | 3.2 |
| Canada | Nova Scotia, Bay of Fundy, Sackville | 2(a2) | site I | Pseudomorph | 41.6 | 1.57 | 0.56 | 1.6 | 11.2 | 3.8 |
| Canada | Nova Scotia, Bay of Fundy, Sackville | 2(a3) | site I | Concretion | 165.2 | 2.94 | 3.16 | 44.6 | 14 | 35.9 |
| USA | Alaska, Utqiavik (Barrow), Isatkoak Lagune | [7] | | Pseudomorph ‡ | 1.6 | 0.87 | 0.01 | 1239 | 3.2 | 0.3 |
| Germany | Nordfriesland, Wadden Sea | 2(b2) | Skt. Peter-Ording | Pseudomorph | 24.9 | 1.08 | 0.17 | 0.6 | 4.6 | 2.3 |
| Germany | Nordfriesland, Wadden Sea | 2(b2) | Halligen/Sudfall | Pseudomorph | 24.8 | 1.04 | 0.2 | 0.8 | 7.6 | 2.5 |
| Germany | Nordfriesland, Wadden Sea | 2(b2) | Kating | Pseudomorph | 19.2 | 0.84 | 0.59 | 0.2 | 5.1 | 3.2 |
| Germany | Nordfriesland, Wadden Sea | 2(b2) | Eiderstadt | Pseudomorph | 28.9 | 0.94 | 0.95 | 0.9 | 12.6 | 3.8 |
| Germany | Nordfriesland, Wadden Sea | 2(b2) | Tönning | Pseudomorph | 43.3 | 1.49 | 6.79 | 3.5 | 4.1 | 14.8 |
| Germany | Nordfriesland, Wadden Sea | 2(b2) | Jeves | Pseudomorph | 28.7 | 1.05 | 0.13 | 0.1 | 19.2 | 4.2 |
| UK | Scotland, Glasgow, River Clyde estuary, Cardross mudflats | 2(c2) | | Pseudomorph † | 108.5 | 2.68 | 2.35 | 2.6 | 4.8 | 24.1 |
| Netherlands | Noord-Holland, Den Hoorn | 2(b2) | Bobeldijk | Pseudomorph | 29.5 | 0.92 | 0.23 | 0.5 | 8.0 | 2.4 |
| USA | Alaska, Seward Peninsula, Sarichef Island, Shishmaref coast | 2(d1) | | Pseudomorph | 88.0 | 2.18 | 0.68 | 4.8 | 5.0 | 14.5 |
| Russia | Kola Peninsula, White Sea, Terski coast, Olenitsa | 2(e1) | | Pseudomorph * | 74.3 | 2.18 | 0.32 | 1.2 | 1.4 | 17.5 |
| Russia | Kola Peninsula, White Sea, Terski coast, Olenitsa | 2(e1) | | Bivalve * | 3.0 | 1.78 | 0.07 | 2.5 | 2.5 | 0.9 |
| Russia | Kola Peninsula, White Sea, Terski coast, Olenitsa | 2(e1) | | Mussel * | 6.0 | 1.49 | 0.23 | 0.4 | 4.4 | 1.2 |

* Published in Vickers [17]; † Published in Shearman and Smith [9]; ‡ Published in Schultz [10].

*Samples and Sites*

This paper builds upon earlier works [13,14,19,20] and presents additional data and descriptions for various locations. The sites investigated are Dutch/German Wadden Sea, Scottish pseudogaylussite, Alaskan Kool Hoot, Russian White Sea Hornlets, and Canadian Fundylite. The use of different names in different locations reflects the initial lack of realization that these pseudomorphs shared a common parent mineral. The available site information is sparse and incomplete, reflecting the limited published data, loss of sites, and the inaccessibility of remaining sites. However, the use of museum specimens has allowed for the collection of stable isotope, $^{14}$C dating, morphological, and petrographic data.

The most extensively studied site is Olenitsa on the Kola Peninsula in Russia (Figure 1e), which has been well described in publications [21–23]. Abundant samples from this site are available for analysis. Additional $^{14}$C dating and information on preserved sediment structures, not emphasised by Geptner, have been obtained. The site at Shishmaref on Sarichef Island, located on the Alaskan Chuchov coastline, has limited published information from 1953 [24]. However, samples and observations have been made available by hydrologist Paul Burger from the US Park Survey, who collected samples from the slushy mud at Shishmaref inlet. The findings confirm that the pseudomorphs are glendonite and provide an age for the site. The River Clyde site in Scotland is only known from historic publications best shown by MacNair [25], where samples retrieved by dredging mud from the river are presented with a petrographic slide and some sample descriptions. Additional $^{14}$C dating has been performed on samples from the Copenhagen Museum of Natural History and thin sections from a sample obtained by Douglas Shearman from the Huntington Museum. The sites along the Wadden Sea coastline in the Netherlands and Germany are primarily known from historical references and museum collections [13]. The site at Bobeldijk, close to Den Hoorn, the Netherlands (Figure 1b), was briefly rediscovered in 2003, while the others are only mentioned in references [13,14,20,21,26,27]. The Hamburg Mineralogisches Museum holds a unique and extensive collection of samples from original sites, often by the first authors in the 19th century. The museum allowed the examination of petrology and $^{14}$C dating on a few selected samples, contributing to the collective knowledge of the topic. The site at the Bay of Fundy in Canada is well documented (Figure 1a), providing a site description and $^{14}$C dating of wood stumps covered by tidal mudflats [28]. Local collectors Dana Morong and Donald Hattie have recovered pseudomorph samples from the original site, enabling the presentation of petrography and $^{14}$C dates for these samples. A key observed from Figure 2 is that the samples from each site are very uniform in size and shape, yet often not preserved with high detail [16]; therefore, the figure aims at showing morphological conformity and not crystallographic detail.

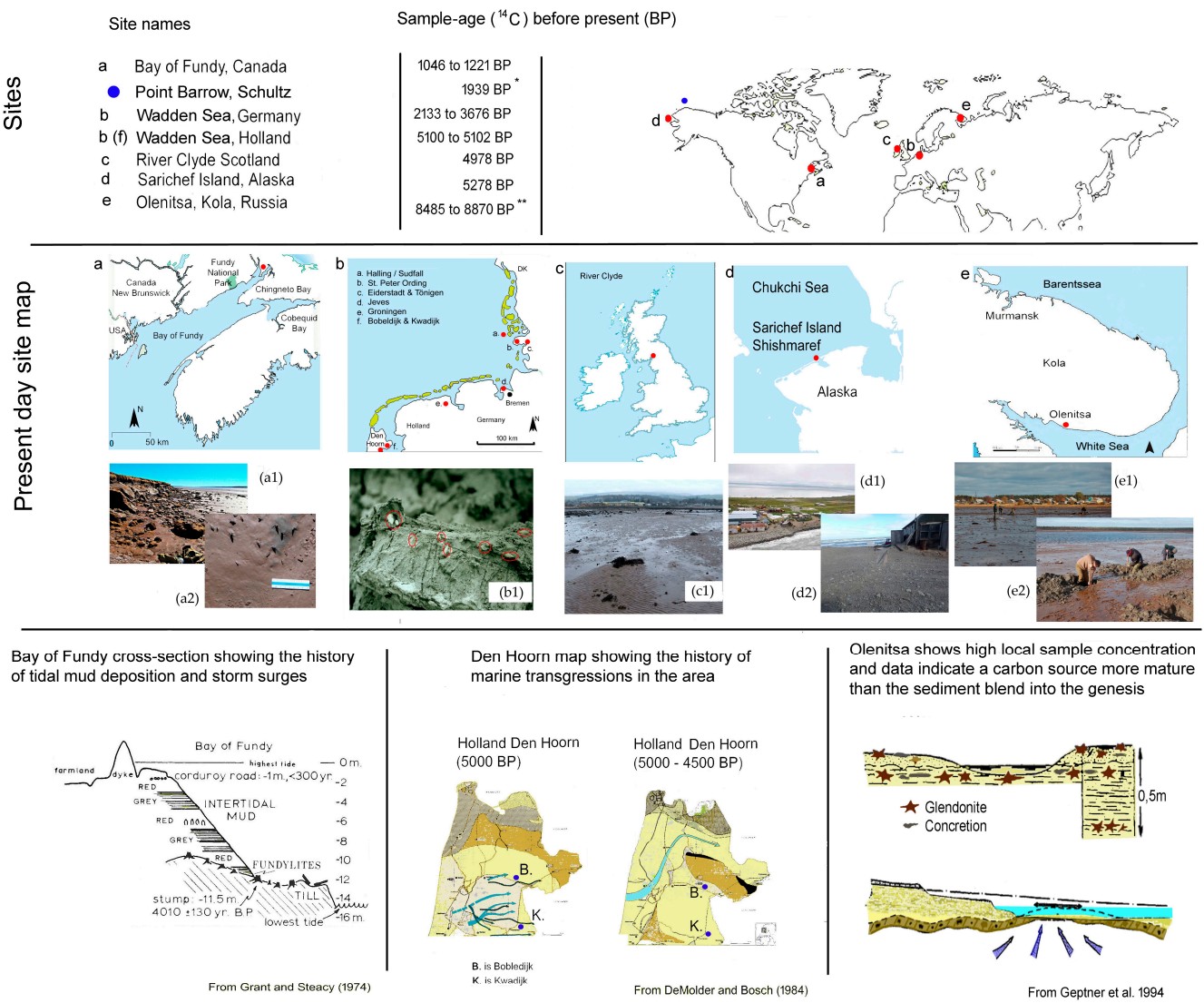

**Figure 1.** Photo (**a1**,**a2**) from the Bay of Fundy in Canada by Don Hattie and Dana Morong. Photo (**b1**) from Bobeldijk in the Netherlands by Jan de Jong. Red circles indicated to glendonites are shown in Figure 2b. Photo (**c1**) from Cardoss Sands at River Clyde in Scotland from Lairich Rig. Photo (**d1**,**d2**) from Shishmaref in Alaska by Paul Burger. Photo (**e1**,**e2**) from Olenitsa in Russia by Alexia Rogov. * [10],** U/Th suggest 4.1 ± 0.4 cal. Ka BP [23].

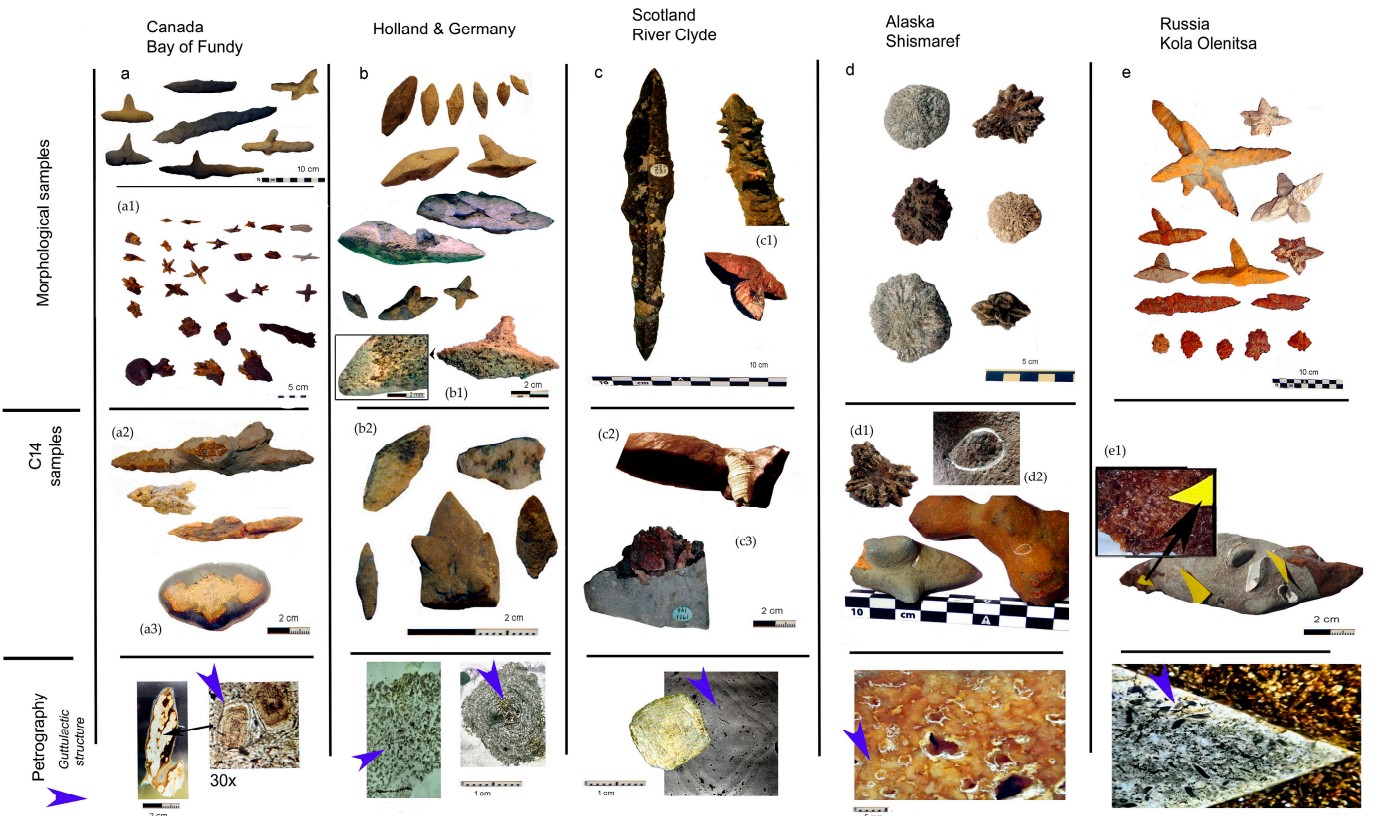

**Figure 2.** Samples are shown in chronological $^{14}$C order. The samples are all similar in morphology and size regardless of origin. All samples have a calcite matrix with a mesh of primary guttulatic calcite followed by later generations of calcite. Some, but not all, are fully cemented. (**a**) is Bay of Fundy site 1 samples, (**a1**) are site of 2 samples (photos by Donald Hattie). As the pseudomorphs hold no detail, we show what is observable given as a general morphology. (**a2**) the Bay of Fundy, $^{14}$C samples sites 1 and 2, where (**a3**) is the concretion sample. (**b**) are from Wadden Sea sites in Germany from Skt. Peter-Ording, Halligen and Tönning. (**b1**) displays the general morphology of all Wadden Sea samples with a more dens rim/crust covering a mesk of minute mm. large crystals. (**b2**) are $^{14}$C samples. (**c**) are River Clyde samples, with (**c1**) displaying rarely observed parasitic growth indicating more than one sequence of crystallisation, and (**c2**) is the $^{14}$C measured snail and concretion. (**c3**) is the concretion with a glendonite encased. (**d**) are samples from Shishmaref, where (**d1**) are the 14C samples with (**d3**) showing detail to the $^{14}$C measured clam encased in concretion also holding glendonite. Order cannot be stablished Sample (**e**) is from site Olenitsa, with (**e1**) being the $^{14}$C sample holding both pseudomorph, bioclast and concretion in that order.

## 2. Methods

### 2.1. $^{14}$C Accelerator Mass Spectrometry (AMS)

Dating of the carbonates was undertaken on bulk pseudomorph by $^{14}$C AMS at the Institut for Fysik og Astronomi, Århus Universitet, using an HVE 1MV accelerator. The measured dates for marine carbonate samples are corrected for the ocean reservoir effect to be comparable to contemporaneous terrestrial material by subtracting the ocean reservoir age from the conventional (measured) $^{14}$C age. Calibrated ages in calendar years have been obtained from the marine model calibration curve (Marine 13) using the Oxcal v4.1 programme with the probability method of Reimer [29]. The marine model accounts for the smoothing in the world ocean of the sharper variations in the atmospheric calibration curve. If not specifically stated otherwise, the local deviation (Delta-R) from the average world ocean model is assumed to be zero, corresponding to a standard reservoir age of 400 years. The $^{14}$C ages are reported in conventional radiocarbon years BP (before present = calendar year 1950) in accordance with international convention [30].

*2.2. Inductively Coupled Plasma Optical Emission Spectrometry (ICP-OES)*

Minor element analyses were undertaken on powdered, bulk-dried, transformed ikaite (now calcite) using an Agilent 5110 VDV ICP-OES at the Camborne School of Mines, University of Exeter, following the method detailed by Ullmann [31]. The minor element data are expressed as ratios to Ca and calibrated using certified single-element standards mixed to match the chemical composition of the analysed samples. The precision and accuracy of the analyses were measured and controlled by interspersing multiple measurements of international reference materials, JLs-1 and AK, and quality control solution (BCQ2).

## 3. Results on Petrography as Indication of Recrystallised Ikaite (Glendonite)

*3.1. Bay of Fundy*

The pseudomorphs known as Fundylite consist of numerous small guttulatic calcite granules, each measuring 0.2 mm. Like other pseudomorphs, a surface calcite rim has prevented sediment from entering the internal porosity of the pseudomorph. Recrystallisation may have occurred in two phases, as later cement generations exhibit typical calcite petrography. The pseudomorphs show a variety of textures, including partially filled voids, cemented meshes, and secondary concretions. Recrystallised ikaite in the earliest generations suggests that ikaite was the precursor mineral for the pseudomorphs in the White Sea region.

*3.2. Wadden Sea*

Eiderstadt, Halligen, and St. Peter-Ording samples consist of guttulatic calcite crystals in a porous mesh throughout the pseudomorph cavity. The samples have a solid calcite rim and an internal mesh of prismatic calcite with a distinct guttulatic morphology are present. The calcite contains 3–5% Mg [26]. Only one cement generation is recognised, and secondary concretionary calcite is rarely observed. The pseudomorphs often appear to float in the surrounding sediment.

*3.3. River Clyde*

The River Clyde pseudomorph is fully cemented and exhibits petrography similar to Svalbard pseudomorphs [32], with a calcite mesh arranged in layers as rim cement, botryoidal calcite, and guttulatic prismatic calcite crystals. Secondary concretions are rarely observed, and pseudomorphs often appear to float in the sediment.

*3.4. Sarichef Island*

The Sarichef pseudomorph interior is nearly fully cemented by a semi-porous mesh of guttulatic calcite crystals. Some samples are encased in secondary concretionary calcite.

*3.5. White Sea*

White Sea pseudomorphs comprise large prismatic guttulatic calcite grains overgrown by one or more generations of later non-guttulatic calcite. Pseudomorphs show a variety of textures, including partially filled voids, cemented meshes, and secondary concretions.

## 4. Isotope Data, Radiometric Dating, and Geochemistry

*4.1. Bay of Fundy, Canada*

The bay is both a geopark, a national park and a world heritage site, because of which there is copious information available for the region, and most importantly, from the perspective of this study, the glendonite site still exists. A major evaporite diapir comprising Mississippian age salts (Windsor Group) has its crest about 2 km north-northwest of where the fundylites are found. The nature of the rocks underlying the fundylite-bearing horizons is unclear, but it is most likely a Pennsylvanian-aged fluvial sandstone and conglomerate that flank the salt structure.

The Bay of Fundy is renowned for its extreme tides (up to 13 m) and cold waters, resulting in the deposition of organic-rich tidal mudflats. The pseudomorphs occur prefer-

entially in sediments near riverbanks and the margins of salt marshes. This, along with a slight coastal topographical high, suggests that the Cumberland Basin was a lake during part of the Holocene epoch. These sediments were later covered in salt marsh sediments as sea level rose [33]. Today, the silty till plain and hummocky ground moraine glacial deposits attest to a glaciated past [33]. Salt marshes form on submerging coastlines when the sedimentation rate exceeds the rate of land submergence [34]. The Chignecto Basin, located at the head of the Bay of Fundy, is an example of a submerging coastline, with sea level rising approximately 3 mm per year. Today, the middle of the continent is experiencing an isostatic rebound, and the margins are submerging, resulting in a transgression [35].

In situ, glendonite-type pseudomorphs were first reported from the tidal mudflats of the Bay of Fundy, Canada, and the crystals were named fundylite [15]. The fundylite-bearing exposure is very limited in both vertical and horizontal extent. There are several red or grey tidal silty mud deposits with a total thickness of 12 m. The fundylite-bearing mud is only exposed at the foot of the levee and, most importantly, overlies tree stumps dated using $^{14}$C dating to be 4050 BP old, as shown in Figure 1 [28]. Local collectors Don Hattie and Dana Morong have surveyed the site, retrieved some 32 samples included in this paper, and reported informally that there are more fundylite outcrops in the area. At Site I pseudomorph dates at $^{14}$C 1046 +/− 22 and 1059 +/− 27, with concretions being younger at $^{14}$C 762 +/− 30. Site II dates a little older at $^{14}$C 1221 +/− 26. Fundylite is partially or fully enclosed in concretionary calcite. The samples from the original site are slender forms encased in concretions and are longer than the bladed form depicted in Figure 2. The small and the elongated pseudomorphs are like those from the river Clyde. Although the fundylite is porous, the host sediment has not penetrated it, nor has the concretionary calcite cemented this porosity. Key elements such as monoclinic affinity, a concave/convex structure, and pseudo-pyramid pinacoid prismatic faces can be identified, even with poor preservation, using high-resolution 3D photometry.

Primary sediment structure is preserved in the concretions. Bioturbation and trace fossils observed in the sediment are not preserved in the concretions. However, tidal mud sedimentary structures that are not preserved in the clay are present in the concretions. From the orientation of preserved sediment structures in the concretions, it appears that the presumed ikaite formed at random crystal orientations in the sediment. This is consistent with an environment where water moves through and out of the sediment. The concretions, however, have petrified roots and small fish bones embedded in the material. No marine bioclasts were observed in the concretions. One sample is attached to what appears to be a drop stone.

### 4.2. Wadden Sea, the Netherlands and Germany

Geographically, the region is well-known and uses the known sample ages. We can work out the climate at the time of precipitation, something not achieved in earlier papers. There appears to be a correlation of glendonite with winter storm sediments in mature coastal lowlands. This observation shows that even as the sites are further apart, they hold relatively close ages. After the first discovery of pseudomorphs in Thuringia at Sangerhausen in 1826 by Freiesleben [36], and named pseudogaylussite in 1836 by Breithaupt [37], there followed twelve further reports of pseudomorphs found during excavation of dykes, dams, and bridges in the vicinity of the Dutch Wadden Sea coastline [38]. However, given the extensive excavation for dikes, dams and draining since 1841, remarkably few pseudomorphs have been reported from the Dutch and German North Sea coastline [27,39]. Some site reports provide summaries of sediment type and location but not enough detail to permit a modern interpretation of depositional environment or diagenesis. Only a few, such as Skt. Peter-Ording has recent finds of glendonite concretions, yet none of these concretions are in situ [27]. Older references [26,40] and the more modern [27,38] describe marine marls and silty clays exposed during excavation for foundations, dikes, or dams, with no record of the structure of the sediment they were found in. At the Hamburg Mineralogical Museum, there is one sample from Halligen with a sedimentary structure

that strongly resembles the sedimentary texture in samples from White Sea Olenitsa with laminations and micro-hummocky structures (cf. Einsele (1992) [41] specifically Einsele Figure 7(1d) after Bayer [42]). These features are consistent with rip currents and small channel fillings generated beneath the ice. Hamburg Mineralogisches Museum has samples from historical sites 10–20 km apart. This enabled us to cross-reference the $^{14}$C ages, showing in the German part of the Wadden Sea that the events and the carbon sources are similar. Viewing the pairs, e.g., Kating/Eiderstad, Halligen/Skt. Peter-Ording (Figure 3) would suggest a possible link to a series of events in the development of the Bronze Age North Sea Coastline, whilst bearing in mind that the sites have become exposed due to modern-day erosions from storm floods over the past 500 years.

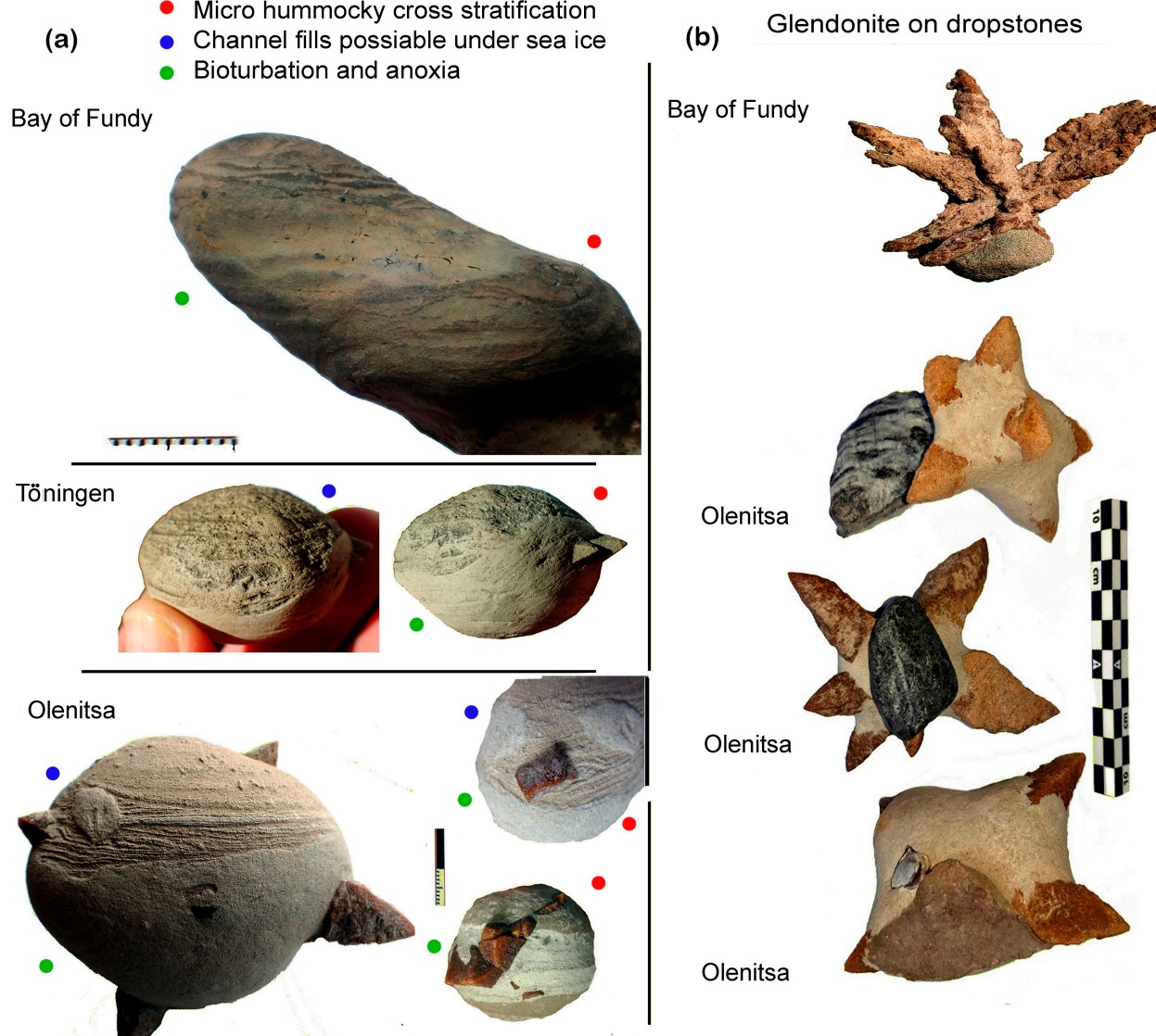

**Figure 3.** (**a**) Sediment structures from three sites suggest high-energy depositional environments and bioturbation. (**b**) Drop stones, rounded and rugged, pebble to cobble size, are interpreted as ice-rafted. Drop stones may be a consequence of wintertime storm surges during transgression events.

This provides sample ages for the sites near Den Hoorn (Noord-Holland), situated 15 km apart, with Bobeldijk at $^{14}$C 5203 ± 32 BP and Kwadijk $^{14}$C 5100 ± 34 BP. For the German sites, the samples consist of two pairs. One is from Halligen and Skt. Peter-Ording is 20 km apart, Hallig at C$^{14}$ 3666 ± 35 BP and Skt. Peter-Ording at $^{14}$C 3676 ± 30 BP. The

other is Kating and Eiderstadt/Tönning, which are 20 km apart, Kating at $^{14}$C 2872 ± 40 BP and Eiderstadt at $^{14}$C 2911 ± 22 BP.

Our study of the Hamburg Mineralogisches Museum's vast collection of pseudomorphs confirms that all non-weathered samples conform to the morphological types described yet add little to the comprehensive descriptions [16]. A few samples from Tönning are remarkably well preserved as they were retrieved from excavations according to Rose reported in 1841 [40]. Glendonites with blades and small clusters have been found in the German part of the Wadden Sea, but no rosette pseudomorphs have been found. Rosettes have, to date, only been found around Groningen (the Netherlands), but these have not been included in this study as no samples are available. A site in Dithmarschen (Germany) is also not included as no find details are recorded.

*4.3. River Clyde, Scotland*

The sediments record multiple stages of glacio-marine deposition, and there has been significant uplift since the end of the last glaciation. Interbedded fluvio-glacial and marine deposits indicate a time of alternating transgression and uplift [43]. A later marine transgression flooded coastal peat bogs [44].

The samples described as jarrowite were dredged from mud in the River Clyde, Scotland, while testing machinery. Consequently, they have never been observed in situ, only known as concretions and pseudomorphs being dredged up from bioclast beds [9,25]. In the Natural History Museum collection in London, there are no concretions, but in the Copenhagen Museum of Natural History, there is a concretion fragment from the same locality around a fossilised worm tube burrow. There are many other accessible samples in the Natural History Museum in London and the Museum of Natural History in Copenhagen, though only the latter has allowed sampling. Museum samples investigated are listed with references. A marine gastropod enclosed in concretionary calcite has been dated at $^{14}$C 4978 ± 25 BP, where the pseudomorph dates at $^{14}$C 5693 ± 31 and the concretion at $^{14}$C 5693 ± 31.

Samples in the Natural History Museum (London) are all the elongated bladed types, while the samples in the collection of the Natural History Museum in Copenhagen are morphologically much more diverse, including bladed pseudomorphs with a diamond-shaped cross-section or an elongated cross-section. Concretionary calcite around the pseudomorphs preserves bioturbated sedimentary structures. One unusual sample displays aggregated parasitic growth, with many small crystals forming on a larger one. This phenomenon is commonly encountered in other minerals forming in sediments but is rare amongst glendonites. The concretion sample has a cluster or rosette-like morphology, which has been impossible to identify. One pseudomorph encloses a marine gastropod that has been invaluable for determining age and depositional environment. As there is very little material to interpret, the study of concretions can only add that there is evidence of bioturbation preserved in the concretions, with no evidence of dropstones.

*4.4. Shishmaref Inlet, Sarichef Island, Chukchi Sea, Alaska*

Sarichef Island is composed of fine-grained silty-sand permafrost, most likely wind-blown deposits formed as dunes when global sea levels were much lower. Marine bivalves and gastropods are preserved in a life position in the sediment. The area encompasses 2.8 square miles of land and 4.5 square miles of water. Shishmaref (in Inupiaq: Qigiqtaq) is an inlet within the Bering Land Bridge National Preserve. Sarichef Island is part of the Holocene barrier island complex running along the coast of Alaska, facing the Chukchi Sea, just north of the Bering Strait and five miles offshore from the mainland. The island is very vulnerable to erosion as it is made entirely of soft sediment. The samples reported in 1953 were found washed up on the shore, where locals had, for many years, observed such "stones" washed up on the beach or pushed landwards by ice after storms [24]. Hydrologist Paul Burger has, in recent years, collected around 15 samples herein described for the first time.

Some samples are enclosed in concretions with marine fossils as inclusions and partially to fully bioturbated sedimentary structures preserved. The bivalve dates at [14]C $5278 \pm 38$ BP, showing a more realistic age of the sediment and genesis. Whereas both the clam bearing concretion dated at [14]C $10702 \pm 46$BP and included pseudomorph at [14]C $9853 \pm 49$, along reference pseudomorphs dated at [14]C $9653 +/- 43$ BP and [14]C $10162 \pm 47$. This indicates that the carbon source is of a more mature origin than the sediment. Most of the samples are rosettes with multiple points like those from Olenitsa. Three of the samples have an uncommon, flattened rosette form. One sample is the clustered form. Two samples have a convex and a concave side. Some pseudomorphs have a "staircase" of prismatic faces on the convex side and a slightly distorted tip conforming to the defining morphology [9]. The pseudomorphs are in concretions in partially to fully bioturbated sediment. As at Olenitsa, the sediment is silty sand with marine bivalve fossils preserved in a life position. The concretions and pseudomorphs are like Fundylite and the European sites in that the pseudomorphs are porous with only partial secondary cementation encased in fully cemented concretions.

### 4.5. Olenitsa, White Sea, Russia

Pseudomorphs occur on the remote Terski coast, where the Olenitsa River enters the White Sea [21–23], and marine Holocene deposits are widespread. Figure 1 shows a model where Holocene sediments from older deposits underlying the moraine are covered with moraine and Holocene marine deposits, followed by alluvial sands and pebble beds. The site shows the supposed roof of eroded marine Holocene deposits with the pseudomorph and concretion-bearing sediments overlying littoral sediments with a supposed upward flow of carbon-containing gases, as shown in Figure 1. [21]. Three C[14] dates from pseudomorphs are 10.14 Ky to 9.44 Ky [21]. In 2020, Madeleine Vickers published cold environment-related clumped isotope data from the site [17].

Our [14]C ages, ranging from 8870 to 8485 BP, are a bit younger than former measurements suggest. They were obtained from samples where pseudomorph and marine bioclast are embedded in the same concretion. The oldest dates are for the pseudomorph ([14]C $8870 \pm 61$ BP) and a concretion-encased marine bivalve ([14]C $8835 \pm 49$ BP). The youngest age is for a mussel ([14]C $8485 \pm 65$BP) and the surface of a concretion ([14]C $8559 \pm 52$ BP0 [17]. Differences in ages between *Mytilus* and *Astarte/Mya* may be attributed to the fact that the latter taxa lived in the sediment (infaunal), capturing carbon from a mixed source, while *Mytilus* is an epifaunal bivalve taxon. Geptner [21] notes that pseudomorphs occur in concretions that enclose marine bivalve bioclasts, as shown in Figure 3. The concretions encase the original sediment and sediment structure, also shown in Figure 3, which consists of a sharp aerobic–anerobic boundary, observed as a sharp change from bioturbated to un-bioturbated sediment. Small channel fillings with reworked mud lumps indicate submarine erosion and sediment lamination, and micro-hummocky structures occur in the silty sand, like those in the Wadden Sea samples [40]). The similar size of the bivalves and the life position of the *Mytilus* is consistent with them having been buried in a storm event. Using the extensive photographic record of single crystals and crystals in concretions [21,22], we have observed that the pseudomorphs have the concave and convex form consistent with the features defined as requisite for glendonite identification [16], and they fall into the three categories of rosettes, clusters and bladed [16,45]. The suspected drop stones shown in Figure 3 are thought to be such as they are very variable in size and come as both angular and rounded. If they came simply from the nearby beach, all would be expected to be rounded pebbles.

## 5. Discussion Isotopes and Mineralisation Event

Two different models for ikaite formation have emerged from work at Olenitsa [21,22] and the Isatkoak site [10]. The Olenitsa model involves a shallow meteoric seep in coastal regions during transgression, while the Isatkoak model requires a single forceful event, such as a storm surge.

### 5.1. Bay of Fundy

The pseudomorph yielded a $C^{14}$ date of 1046 +/− 22 BP. Measured $\delta^{13}C$ ranges from −17.59 to −19.9 (‰ V-PDB) and a $\delta^{18}O$ value of 0.22 (‰ V-PDB), consistent with a mixing of seawater DIC and organic matter diagenesis. Campbell [46] reported that the pseudomorph-bearing mudflat layer is deposited over old tree stumps $^{14}C$ dated at 4050 BP. However, It is not established when the mudflat was deposited [28]. The $\delta^{13}C$ data are consistent with methane released from anoxic settings into overlying marine sediments.

### 5.2. Wadden Sea, the Netherlands and Germany

The $\delta^{18}O$ and $\delta^{13}C$ values point to a seawater source with mixing of inorganic and biogenic carbon. None of the data indicates an influx of meteoric waters from older fault systems, with $\delta^{18}O$ ranging from −1.06 to −1.75 (‰ V-PDB) and $\delta^{13}C$ ranging from −24.13 to −13.47 (‰ V-PDB) [46].

A precise description of the German Wadden Sea has not been located, but there is a thorough description of Den Hoorn in Noord-Holland, as shown in Figure 1 [47]. The coastline shifted quite a bit between 16,000 BC and 7000 BC, and the lowlands known as "Doggerlands" between England and Europe slowly flooded. However, our sites are younger than these events. To date, little data has been found for the Eiderstadt area. However, a former PhD student at Hamburg University, A. Cordua (pers. comm.), states that a 3 m deep drilled core named GeoHH—SH01 retrieved on the Eiderstadt peninsula north coast had an embankment age of 1862 AD. At 2.1 m depth, the $^{14}C$ age was 900 BC (2920 BP), where the sediment was fossil marsh. A map made in 1648 depicts the site as several km from the coastline, though it should be noted that extreme storm floods of 1362 and 1634 had moved the coastline many kilometres inland and eroded much of the Wadden Sea lowlands. In summary, all Wadden Sea sites formed in the hinterland of the coastline. For the sites at Bobeldijk and Kwadijk, we have found references to the evolution in the coastal terrain $^{14}C$ [47]. Palaeogeographic maps for the Holocene sediments around Den Hoorn are consistent with an age >5000 BP, showing the Wadden Sea area was influenced by washover events [47]. Glendonite only occurs in areas of steady sediment accumulation, where under topsoil and marine deposits lay Pleistocene moorlands [47]. Glendonite is absent from channel-filling silty sands. The Holocene transgression phase, Calais III (5300 to 4700 BP), corresponds to the expansion of the marine area in the sandy channel deposits, followed by Calais IVA age (5000 to 4100 BP) [47].

From palaeo-environmental studies, the Early Bronze Age climate was warmer than today. This warm period was followed by a colder, more humid period, correlated with shifts in mean high-water levels and $^{14}C$ -dated archaeological findings from around 700 BC [48,49]. This shift has been linked to changes in settlement patterns in the Wadden Sea area, leading to an expansion of salt marsh settlements during the following centuries [50,51]. The subsequent temperature drop is purported to have resulted in increased storm surges. However, [50], little evidence has been presented that supports such a mechanism during the Bronze Age temperature shift. In the northernmost areas of the Wadden Sea, these changes have also been correlated with large areas of fine sand being exposed and shifted from around 700-1 BC [52,53]. Similar shifting sand was absent in the Dutch and German coastal areas due to different soil compositions [52,53].

Accepting that the early Bronze Age climate was warmer or like modern temperature levels, it would be fair to evaluate the groundwater temperature using the present-day annual groundwater temperature in the Wadden Sea, which is 9 °C at a depth of 10 m. The $\delta^{18}O$ and $\delta^{13}C$ values point to the source being seawater with mixing of inorganic and diagenetic carbon. None of the data indicate an influx of meteoric waters from older fault systems, with $\delta^{18}O$ ranging from −1.06 to −1.75 (‰ V-PDB) and $\delta^{13}C$ ranging from −24.13 to −13.47 (‰ V-PDB) [46].

### 5.3. River Clyde, Scotland

For biogenic carbonate (gastropod), the $\delta^{13}$C value is 2.49 ± 0.05 (‰ V-PDB), and $\delta^{18}$O is 1.43 ± 0.05 (‰ V-PDB), consistent with precipitation from seawater with a contribution from inorganic carbon [44]. $\delta^{13}$C values measured from a pseudomorph are −24.86 (‰ V-PDB) and $\delta^{18}$O value of −0.46 (‰ V-PDB). The carbon data indicates mixing diagenetic and seawater inorganic carbon [42]. There is no evidence for meteoric water or burial diagenesis. Samples from the relatively nearby Jarrow Lake on Tyneside have C and O stable isotope ratios like those for the river Clyde site (Douglas Shearman pers. comm.).

### 5.4. Shishmaref Inlet, Sarichef Island, Chukchi Sea, Alaska

Only reworked samples are found at Shishmaref, which precludes precise age dating of the sediment they were deposited in. The postglacial flooding of the Bering Strait is dated at 11,000 Ky BP [54]. After that, the sea level rose rapidly but flattened around 7000 Ky BP. Around 5000 BP, the sea level would still have been 5–10 m below the present-day level. The bivalve in a glendonite bearing concretion shown in Figure 2 as 2.9a yields an age of $^{14}$C 5278 ± 38 BP, which corresponds well with the events at Shishmaref on present-day Sarichef Island.

### 5.5. Olenitsa, White Sea, Russia

A comprehensive account of the mineralogy and isotope data for the White Sea pseudomorphs was provided by Geptner in 2014 [22] with clumped isotope data for the three different materials: marine bioclasts, concretions and glendonite. This data yields a calculated palaeotemperature of +2 to +5 °C, which lies in the range for naturally occurring ikaite [4,17]. The $^{14}$C ages are confined to 8870 to 8485 BP, with the oldest being the pseudomorph at 8870 ±61 BP and a marine bivalve encased in a concretion date to $^{14}$C 8835 ±49 BP. The youngest is a mussel at $^{14}$C 8485 ± 65 BP, and the surface of a concretion at $^{14}$C 8559 ±52 BP. Carbon and oxygen isotopes from the bioclasts indicate marine conditions. The concretion stable isotope data indicates mixed sources and the pseudomorph anoxic biogenic diagenesis. In 2022, Vasileva published a high precision Th/U age placing the glendonite pseudomorphs formed around 4.1 ± 0.4 cal. Ka BP [23]. To interpret the discrepancies in measured dates, we examined the regional palaeogeography. Peat accumulation on the Kola Peninsula started circa $^{14}$C 8500–7500 BP, where pollen from the tree *Pinus sylvestris* indicates it reached its present northern limit on the peninsula by 7000 BP, while $^{14}$C 6000–5000/4500 BP was a time of maximum extension of Birch forest tundra to the Barents Sea shoreline [55]. A first indication of loss of (or deepening of) permafrost is the presence of burrowing rodents, which indicates that topsoil was no longer in the permafrost zone by 7500 BP [56]. This loss of permafrost will have altered the hydraulic flow regime. Carbon and oxygen isotopes from bioclasts indicate marine conditions. The restriction of glendonites to Olenitsa may be linked to either local methane degassing or seeping of mineral-enriched waters from higher grounds into the marine mudflats or linked to gas hydrates breaking down. As with other occurrences, the Olenitsa glendonite has lower $\delta^{13}$C values than enclosing concretions.

Given that the age is close to what in GRIP II is referred to as the 8.2 ky cooling event, induced by shifting currents (Bond cycle), it is reasonable to interpret the sedimentary structures as a sandy silty shoal, deposited following a storm over ice-covered water. This would account for dropstones, mussels in life position, and other bioclast debris. The White Sea region shows that the Atlantic molluscs overgrown by glendonites are a particular species that inhabited the White Sea during the Atlantic climate optimum (6300–6900 calibrated years BP) [23], which means that the glendonites are younger than at least 6900 calibrated years BP. It is puzzling that ages obtained from $^{14}$C do not correspond with geological data. The $\delta^{13}$C and $\delta^{18}$O in glendonites and bioclasts differ slightly, reflecting different environments and seasons of precipitation and different mineralogy (bioclasts are aragonitic, glendonites are calcite). In addition, some of the difference could be explained by isotopic fractionation between bivalves and microorganisms that extract C and O (pers.

comm. K. Vasileva). Concretions also contain lithic fragments that are 2–20 cm in size. These can be rough-edged and rounded and appear to be possible dropstones from ice rafts. As concluded in [6] concerning the Olenitsa site, the carbon captured in measured carbonate most likely originates from older sources releasing dissolved carbon as shown by Geptner in 1994 [21], yet as we will discuss all parameters, such as shown in Figure 4, with correlation of Ca/Mg, $\delta^{13}$C and $\delta^{18}$O are not unilateral, indicating that reliability of $^{14}$C derived age estimates may indicate both an age in relation to genesis, but also in relation to carbon origin.

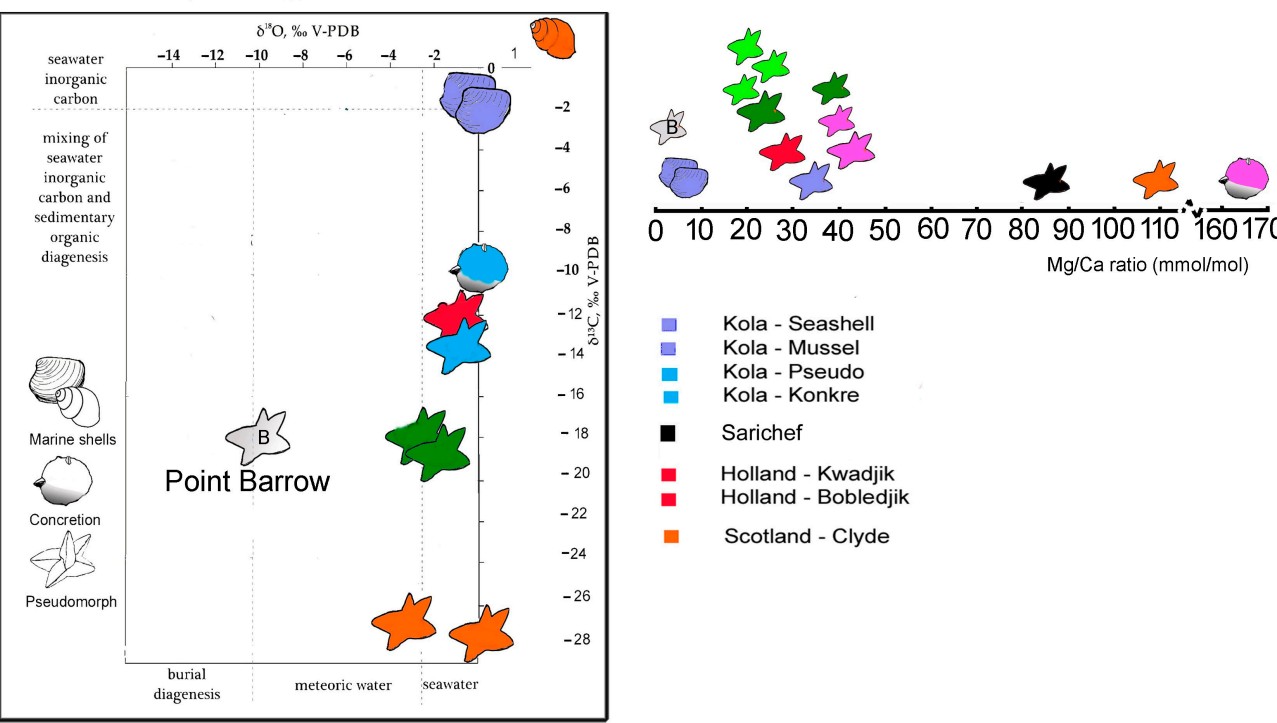

**Figure 4.** Results from Tables 1 and 2 are plotted to show how the analysed samples would indicate a relation to the same type of geochemical genesis. Not only are the sites close in age or geographic position, but also the sites that are widely apart. Point Barrow represents calcite derived directly from ikaite, which is recrystalised out for sediment influence [7], showing that the Mg/Ca ratio is closer to that of marine ikaite [10]. This indicates that this paper's sample calcite was influenced by digenetic porewater fluid. A fact that could explain the linear spread shown in the isotopic plot of ρ$^{13}$C. Olenitsa glendonite data is from [17], and Clyde is from [9]. The plot of stable isotopes are inserted in a display made after Cambell [46].

*5.6. The $^{14}$C paradox*

At Shishmaref, the bioclast $^{14}$C age helps to correlate ikaite precipitation, whilst at Olenitsa, it does not. Our $^{14}$C data from site Shishmaref shows that a pseudomorph dated at $^{14}$C 10702 ± 46 BP is considerably older than the ages derived from bioclasts in the concretion, $^{14}$C 5278 ± 38 BP. At Olenitsa, the Th/U age 4100 ± 400 BP is much younger than our $^{14}$C date range of 8870 to 8485 BP, made using three independent $^{14}$C analyses on bioclasts, for which two have seawater values of $\delta^{13}$C and $\delta^{18}$O. If the bioclast has been diagenetically altered, we should expect the isotopes to be closer to that of the pseudomorph and concretion than seawater. Records of flora and fauna in the Olenitsa region suggest the younger age of Th/U 4100 ± 400 BP correlates better with events than the older age of $^{14}$C 8870 to 8485 BP, as the younger age would have been after permafrost subsided and groundwater circulation began. We suggest that the Olenitsa ikaite formed

in response to the late-Quaternary transgression [57] and coastal erosion that enabled seepage [21], resulting in ikaite formation that incorporated carbon released from older layers [7]. Moreover, in the Bay of Fundy, where we know from Steacy and Grant [28] that the tree stumps (Figure 1) below tidal muds date at 4010 +/− 130 BP and the blue carbon precipitation in the general area are older than 3000 BP (pers. com. Gail Chumura), and therefore predate ikaite $^{14}$C 1046 +/− 22 to 1221 +/− 26 BP, with the concretion being younger at $^{14}$C 762 +/− 30 BP.

In Europe, soil dating indicates that it does not predate the ikaite formation by many years and could contain a carbon source older than the soil. The $\delta^{13}$C and $\delta^{18}$O from the Bay of Fundy could imply older waters originating from fault and diapirs in the area, yet $\delta^{13}$C and $\delta^{18}$O from Europe that holds much the same setting do not show the same tendency. The data are consistent with a local carbon source originating from the sediment, where our illustration in Figure 5 offer some indications to an explanation relating to local conditions, transgression, and land sink. It must be noted that the sites and the cold spells do not match perfectly, it could be that the carbon incorporated in pseudomorphs predates the precipitation of ikaite. The formation of ikaite (the guttulatic petrology indicates the link to ikaite [6,7,10]), normally associated with near-freezing, deep sea or high latitude environments [19], is somewhat surprising in North Sea Bronze Age coastal settings.

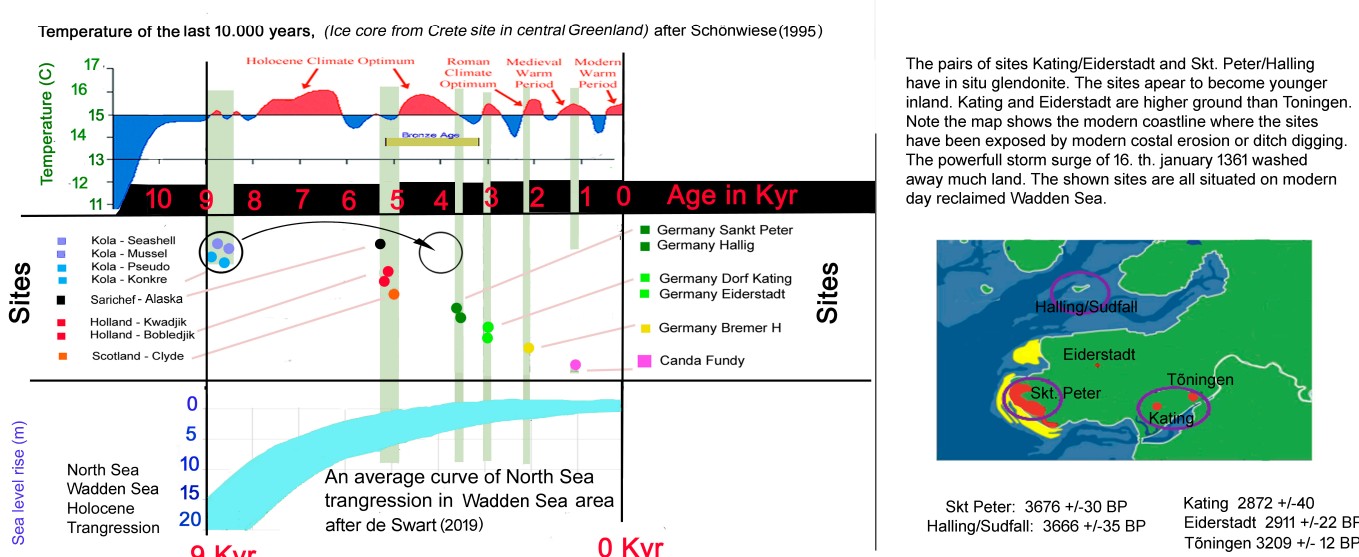

**Figure 5.** Palaeotemperature estimate from Dryas [58] to present and for the Northern Hemisphere and eustatic transgression in the North Sea during the Holocene after Schönwiese [59] and transgression in the Bronze Age after de Swart [60]. The River Clyde has a very complex setting due to significant isostatic rebound, and Shishmaref is of an age where flooding of the Bering Strait closely resembles the curve of the North Sea. Fundy underwent subsidence and marine transgression. If our Olenitsa $^{14}$C dates are incorrect (too old), the Th/U age of 4100 BP is more consistent with other data; this is illustrated by the circle-to-circle arrow [23].

## 6. Conclusions

Our most significant addition to the ongoing investigations of ikaite and recrystallised ikaite is showing that the historic sites referred to as Pseudogaylussite, Kool Hoot, and Fundylite were indeed formed as ikaite, and then recrystallised into glendonite. Yet why ikaite has formed in abundance in limited sections of vast coastal regions is not fully understood. On close inspection, we can see similarities between the places where ikaite/glendonite occur, like shown in Figures 1, 3 and 4. We can demonstrate that the localities recognised today from the Wadden Sea shoreline were previously located in the

hinterland from the current coastal front, as suggested in Figure 5. Isotope and elemental data are consistent with the interaction of spring water and inflowing seawater, leading to mineralisation in regions with significant concentrations of degraded organic material. It appears that a long-running groundwater seep in the Bay of Fundy region caused the development of blue carbon, where the sinking of the land allowed marine life to enter salt marshes and quickly convert carbonate precipitation to ikaite.

The scarcity of glendonite sites implies that ikaite only forms (and is pseudomorphed) under very specific conditions. We have here reviewed data for a wide range of coastal localities from which glendonite pseudomorphs have been obtained, showing that the settings and ikaite formation model for Olenitsa is probably the predominant one, with the mechanism for ikaite formation at Isatkoak, as we offer, from geochemical and isotopic data, being rare. Mg/Ca ratios differ for the two models and provide a guide as to the mode of ikaite formation. They also indicate whether the pseudomorph calcite is purely derived from ikaite or is affected by diagenesis [6]. Glendonites within calcite nodules are often incompletely cemented by secondary calcite and have not been filled with the enclosing sediment, which will influence the geochemical and isotopic results. Previous investigations of bulk material may have failed to correctly identify the diagenetic processes of ikaite forming and recrystallising. We interpret this as due to the formation of a micritic rim around the dissolving ikaite crystal, while incomplete pseudomorph cementation implies that the supply of mineral-enriched porewater was of limited duration following the ikaite precipitation and recrystallisation. In general, we can show that using $^{14}$C as a radiometric dating tool on coastal glendonite offers indications consistent with the site setting yet does not pinpoint the time of ikaite genesis nor its time of recrystallisation into glendonite.

**Author Contributions:** First author B.S. has conceptualized and written the paper supported by second author J.H., an expert on glendonite petrography and by B.v.d.S. The geochemistry has been done by C.V.U. Climate and living conditions were shared by archaeologist M.C.B. All authors have read and agreed to the published version of the manuscript.

**Funding:** This research was funded by (MFO20.2017-004), from the Danish Council for Independent Research Natural Sciences to MV (project DFF-7014-00142).

**Data Availability Statement:** First author and Museum Salling.

**Acknowledgments:** Profound thanks to Jochen Schlutter and the curator, Hamburg Mineralogische Museum has made the study possible as samples are only known from historic collections. Thanks to the Museum of Natural History (SNM) in Copenhagen and Museum curator Xenia Fihl for providing the samples of river Clyde included in this project. Also, thanks to Yma VanDreist, along with Jan and Elly Verkleij from the Netherlands, for providing samples from Bobeldijk and literature. We thank Museum curator Peter Tandy of the Natural History Museum in London for helping in photographing historic river Clyde samples. We thank especially Dana Morong and Don Hattie for Fundy for samples, site information and excellent site images. Hydrologist Paul Burger assisted with samples and background data from Shishmaref. I also extend my thanks to co-authors for a great process, and especially to the academic editors and reviewers for offering good advice making it a greatly improved article.

**Conflicts of Interest:** The authors declare no conflict of interest.

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
