# Peer review of "Transgression Related Holocene Coastal Glendonites from Historic Sites"

_minerals, doi:10.3390/min13091159_

Round 1
Reviewer 1 Report
This manuscript has the potential to provide important data on glendonites found on historic sites and their possible associations with periods of climatic cooling. However, it needs a careful rewriting and reorganization. Currently, the figures and text are disorganized and poorly presented, and it is very hard to follow the author’s arguments. The authors should demonstrate why and how glendonites can be associated with transgression. The authors should be aware that only ikaite and calcite are mineral names. Glendonite, Pseudogaylussite, etc. are names used for describing the ikaite pseudomorph calcite. Therefore, it does not make sense to write that “They contain 88.2 and 86.2% calcite (CaCO3).” (line 151, 155) etc. Also what do the authors mean by Fundylite recrystallization? (line 153) Did calcite recrystallize to a different carbonate? The authors further confuse the situation by writing “that the historic sites refered to as Pseudogaylussite, KoolHoot, Fundylite where indeed formed as ikaite, recrystalised into glendonite ” (line 361). What is Pseudogaylussite, etc.? Is it a pseudomorph, or a historic site? It is unclear what kind of „significant addition” the authors can show in relation to ikaite and recrystallized ikaite. It is a serious problem that the authors improperly use ikaite/calcite/glendonite/pseudomoprh etc. The formula of ikaite and calcite should be added once only (now they can be found in numerous places). another issue is that the improperly use the accepted nomenclature for describing isotopes and isotope data. They should use 14C instead of C14 everywhere and δ instead of the triangle.
I do not agree that the authors could propose a unifying theory for ikaite precipitation. Also, I cannot see a theory in the paper. The authors should specify at what conditions ikaite is metastable (line 41), it would perhaps be better to write that “ikaite transforms to calcite above 7 C.” I am not sure what traditional crystallographic investigation means.
Lot of statements do not have appropriate citations (line 51-55, 68-70, etc) and the references do not match with Minerals requirements (Please compare the reference for line 37 and 39, etc). What is a single-source carbonate? (Line73).
The figures are poorly presented, there are too many panels, and it is unclear what the authors would like to show with them. The panels are way too small and have low resolution. I really do not see why the authors show e.g., panels 1a-5b, etc. I can’t see the marine transgression, long duration seepage etc. It is impossible to see the panels of Fig. 2 and 3. According to title of Table 1, it shows 14C isotope. However, this table shows so many unimportant things that it is extremely difficult to find this data. Similar to the figures it should be simplified.
I suggest a complete rewriting/ spell check as well as reorganization.
It must be significantly improved.
Author Response
Dear Reviewer,
We thank you for your effort and comments on offering us valuable suggestions to improve our manuscript.
We have therefore reflected on your comments as follows:
The script has been reorganized and figures split to follow the contents. As we show that the radiometric dating is not absolute, we refrain from more than suggesting links to cold spells. WE can only clarify for future research that the historic sites formed as ikaite, that at some point recrystalised into glendonite. We can show that our data holds good indication that the ikaite formation, and certainly the recrystalisation occurred under conditions where mature carbon source was involved.
We have taken out the % of calcite, as we correctly do not account for the remaining %.
We have kept the old site names, and name giving as the scope of this paper is to offer future research an update on sites and references. It is within the some odd 30 years of collecting samples and contacts that this paper holds significance as many of the sites we present have not been re-described since first mention, like in Alaska, Scotland, Holland.
We hope that the nomenclature is good now. Thank you for pointing out.
Our misspoke theory is no unifying theory, only another addition to the enigma of ikaite, where we can show the the samples are glendonite types, and that data hint the genesis to be of the same nature occurring within the sediment, and not related to faults or older meteoric waters.
I am sorry that the document no longer correlates to the lines.
Line 37-39, 51-55, 68-70. We have reviewed our citations .
Line 41 corrected
Line 73 - as stated the geochemical reactions appear to be contained within the hoast sediment - a single source.
Regarding detail of figures, we can only agree that higher detail is great if it holds information. Yet we have not be able to make final conclusions with the data available. As only sites like Olenitsa or Fundy are possible accessible and still there, it hampers our ability to conclude on the remaining sites, both regarding settings, and geochemistry. Sites in Germany and Holland are described in relation to landmarks, that are no longer existing, so all we have is museum samples and old references. Over the years all that could have had further information, like local amatures, ditch diggers, Wold Herritage, Universities have been contacted but the effort remained futile apart from whats presented in the paper.
We hope you find the replies and improvement acceptable.
Cheers
Bo
Reviewer 2 Report
The paper is well-written and provides an important data concerning the set of glendonites from historical and partially abandoned sites, and should be published after minor correction
Comments are mainly provided in the attached pdf-file.
There are three general groups of corrections, along with smaller ones:
1) references: both citations in text and references should be carefully checked;
2) figures has numerous small details and hard to read; I propose to split each figures on a few, and enlarge their sizes (examples provided at the end of the pdf)
3) please correct mix of English / German / Dutch names for the same areas (i.e. Wadden Sea, Waddenmeer, Vadenmeer) and institutions
4) an additional table with coordinates for all localities will be very useful, as from the World map the position of studied sites is unclear

Author Response
Dear Reviewer
We thank you for your effort and comments on offering us valuable suggestions to improve our manuscript.
- Citations checked
- Figures split
- English corrected by co-authors
- This is not done as the precise location of historic sites are often given in relation to no longer existing landmarks. The known locations are well represented in their cited articles like Olenitsa and Bay of Fundy.
- We hope that our improvement cover the marked section in your attached pdf. So you may find our paper a better read.
Reviewer 3 Report
For the comments please see the attached file.

Author Response
Dear Reviewer
We thank you for the time and effort on reviewing our paper.
We have greatly improved the manuscripts regarding English and typos.
Figures are split and improved, aiming at offering an overview, as there are no details to show from samples or sites.
Guttulatic is mentioned in line 60.
It would have been so good have better data on Ca/Mg, but as the site sediments are not accessible, we only show our findings based on museum samples and references.
I am sorry that the line numbers no longer are the same.
We have addressed your comments on lines 39, 40.
Regarding geochemical data on ikaite in marine influenced coastal sediments, we cannot clarify beyond our previous papers like 2023 Point Barrow and 2022 History of ikaite and glendonite. It shows that laboratory chemistry is not always compatible to sediments. We find no evidence that could lead us to support the latest model presented by Whiticar et.al 2023. So the enigma of why ikaite forms as it does still stands. We had only offered clarifying observations.
Line 50-54,56-59 - done
Line 72 clarified , 73 reference done
Line 69, 75,85,86 done.
line 102 - 104, the sentence is clarified, so that this observation indicates that the concretion is influenced by more matured carbon, that what the seashell contains.
We hope you find lines 154,164, 160,188,203,206,216,223,272 dealt with.
Round 2
Reviewer 1 Report
I appreciate that the authors revised their ms and changed the figures. Now it is greatly improved. However, I think they can further improve Fig. 1 and 2. The photos of Fig.1 are still too small, and the details are unclear. I suggest the authors magnify these photos and perhaps display them as a separate figure. The same holds for Figure 2. Too many samples are shown for Figure 2, but their details are unclear. I suggest the authors split Figure 2 to more than one figure and magnify the panels to the size of Fig. 3.1. Perhaps it is not necessary to show all the samples.
I suggest the authors reorganize Table 1 and 2. The “Country” and “locality” and “site” occupy large spaces, these should be reduced. I suppose this information can be included into one small box.
The authors should use the accepted nomenclature for describing isotopes and isotope data also in the abstract (line12, line 28). They should use 14C instead of C14 everywhere.
It is unnecessary to give the formula of ikaite, calcite and glendonite more than one. They can still be found in several places (e.g., page line 52, line 68 etc).
Before acceptance I also suggest a through spell check. there are still several unclear/inappropriate sentences. For example (page 1 line 42): “Ikaite (CaCO3●*6H2O) transforms to calcite (CaCO3) above ~7°C, with glendonite now being the name given to this origin for calcite” Too many things are inappropriately mixed in this sentence. The authors could perhaps write that „Ikaite (CaCO3●*6H2O) transforms to calcite (CaCO3) above ~7°C and leaves behind only a few structural relicts”. However, macroscopic calcite pseudomorphs of ikaite, known as glendonites, can be uniquely associated with the ikaite origin “. These sentences clarify that glendonite is a calcite pseudomorph of ikaite, thus unnecessary to give the CaCO3 formula. The first sentence states that only a few structural relicts remain after transformation. The authors can read about these relicts in the paper by Németh, P.; Töchterle, P.; Dublyansky, Y.; Stalder, R.; Molnár, Z.S.; Klébert, S.Z. Spötl Tracing structural relicts of the ikaite-to-calcite transformation in cryogenic cave glendonite. American Mineralogist, vol. 107, no. 10, 2022, pp. 1960-1967. https://doi.org/10.2138/am-2022-8162.
Similar rewriting is necessary for sentence starting with “The Guttulatic” (page 2, line 60-34), etc.
Spell check and rewriting of several sentences are necessary.
Author Response
Dear Reviewer
I hope our response is more readable and you may find our corrections and explanations acceptable.
Cheers
Bo Schultz
